

# Short Communication: Optimizing UAV-SfM based topographic change detection with survey co-alignment

Tjalling de Haas[1], Wiebe Nijland[1], Brian W. McArdell[2], Maurice W. M. L. Kalthof[1]

[1]Department of Physical Geography, Universiteit Utrecht, Utrecht, 3508 CB, The Netherlands.
[2]Swiss Federal Institute for Forest, Snow and Landscape Research WSL, Birmensdorf, CH-8903, Switzerland.

*Correspondence to*: Tjalling de Haas (t.dehaas@uu.nl)

**Abstract.**

High-quality digital surface models (DSMs) generated from structure-from-motion (SfM) based on imagery captured from unmanned aerial vehicles (UAVs), are increasingly used for topographic change detection. Classically, DSMs were generated for each survey individually and then compared to quantify topographic change, but recently it was shown that co-aligning the images of multiple surveys may enhance the accuracy of topographic change detection. Here, we use nine surveys over the Illgraben debris-flow torrent in the Swiss Alps to compare the accuracy of three approaches for UAV-SfM topographic change detection: (1) the classical approach where each survey is processed individually using ground control points (GCPs), (2) co-alignment of all surveys without GCPs, and (3) co-alignment of all surveys with GCPs. We demonstrate that compared to the classical approach co-alignment enhances the accuracy of topographic change detection by a factor 4 with GCPs and a factor 3 without GCPs, leading to xy and z offsets < 0.1 m for both co-alignment approaches. We further show that co-alignment leads to particularly large improvements in the accuracy of poorly aligned surveys that have severe offsets when processed individually, by forcing them onto the more accurate common geometry set by the other surveys. Based on these results we advocate that co-alignment, preferably with GCPs, should become the common-practice in high-accuracy UAV-SfM topographic change detection studies.

## 1 Introduction

Unmanned aerial vehicles (UAVs) are increasingly used for topographic mapping (e.g., Anderson et al., 2019). High-resolution Digital Surface Models (DSMs) can be created from UAV imagery with unprecedented accuracy and low costs from structure-from-motion (SfM) techniques (Westoby et al., 2012; Fonstad et al., 2013). UAV-SfM derived DSMs have therefore been extensively generated for research in a broad range of disciplines and environments (e.g., Lucier et al., 2014; Immerzeel et al., 2014; De Haas et al., 2014; Smith et al., 2016).

Topographic change detection is a powerful tool in geomorphology for linking processes and forcings to rates and patterns of erosion and deposition (James et al., 2012). Effective change detection requires repeated surveys of an area of interest at the relevant geomorphic time scale, sufficient accuracy and precision to resolve changes of the relevant magnitude, and a consistent



reference frame for accurate comparison (Cook, 2017). The recent advances in UAV-SfM techniques have made it possible to meet these criteria at relatively low cost and time demands, resulting in a surge of UAV-SfM based geomorphic change detection studies (e.g., Eltner et al., 2016; Anderson et al., 2019). However, especially for topographic change detection it is of key importance that the differentiated DSMs are both accurate and spatially consistent (Feurer and Vinatier, 2018).

To obtain high-quality DSMs from UAV-SfM, either very accurate camera locations (Turner et al., 2013; Hugenholtz et al.,
2016; Zhang et al., 2019) or precisely located ground control points (GCPs) (James and Robson, 2014; Carrivick et al., 2016), both at cm-scale precision, are needed to georeference the three-dimensional (3D) model and to optimize camera interior parameters and camera positions and orientations. In the absence of both very accurate camera locations and ground control points large errors and distortions may be present in generated 3D models (Carbonneau and Dietrich, 2017; James et al., 2017). In the classical UAV-SfM topographic change detection approach, 3D models are made for each survey and then compared to
quantify topographic change. Recently, however, Feurer and Vinatier (2018) demonstrated that processing aerial images from multiple time steps as a single block in the alignment phase of structure from motion (SfM) processing allows for computing coherent multi-temporal DSMs when using low accuracy GCPs - an approach referred to as "co-alignment" by Cook and Dietze (2019). Cook and Dietze (2019) showed that co-alignment of UAV-derived imagery without ground control results in a nearly identical distribution of measured changes compared to the classical approach using ground control points. They found
that compared to a standard approach without ground control, co-alignment reduces the accuracy level of change detection from several meters to 10 - 15 cm. A drawback of this approach, however, is that it results in high comparative accuracy between the surveys, but that external accuracy is low inhibiting comparison with external data. The recent findings by Feurer and Vinatier (2018) and Cook and Dietze (2019) show that there is great potential in using co-alignment in producing higher-accuracy DSMs for topographic change detection. However, key questions that are left unanswered are: (1) do high-accuracy
GCPs affect the accuracy of UAV-SfM based topographic change detection through co-alignment; and (2) how well does co-alignment perform on datasets consisting of a larger number of surveys.

Here, we compare the accuracy of three approaches for UAV-SfM topographic change detection using nine surveys over the Illgraben debris-flow torrent in the Swiss Alps: (1) the classical approach where each survey is processed individually using GCPs, (2) co-alignment of all surveys without GCPs, and (3) co-alignment of all surveys with GCPs. We demonstrate that
combining co-alignment with GCPs leads to the most accurate topographic change detection, outperforming the classical approach by a factor 4, and that co-alignment strongly improves the accuracy of poorly aligned surveys in the dataset. We therefore advocate that co-alignment should become common-practice in future UAV-SfM topographic change detection studies. This approach can be easily applied and fully automated in SfM software, such as Agisoft Metashape Pro (formerly Photoscan Pro) as demonstrated here, but also in most alternative SfM software packages. Co-alignment requires the alignment
of a much larger number of images, especially when combining multiple surveys. This results in a non-linear increase in processing time (Cook and Dietze, 2019), but this limitation fades with increasing computing capabilities aided by the strong parallelization of SfM methods.



## 2 Study site

We study bed-elevation change as a result of debris-flow activity in the Illgraben torrent in the southwestern Swiss Alps. This
study focusses on the lowest 950 m of the channel (Fig. 1), which has an average gradient of ~4°, an average width of ~25 m,
has steep banks and is incised into an alluvial fan. The alluvial fan surface on the west side of the channel is completely covered
by forest. On the east side of the channel the lowest 450 m is covered by houses, roads, a football field, and some forest, while
further upstream the banks are fully covered by forest.

The Illgraben torrent has a long history of debris flows, and an extremely high debris-flow frequency of approximately 5 debris
flows and debris floods per year since 2000 (McArdell et al., 2007; Berger et al., 2011; Bennett et al., 2014). These debris
flows are generally triggered by intense rainfall during summer storms between May and October, and originate from a
catchment that extends from the top of the Illhorn mountain (elevation 2716 m a.s.l.) to the Rhone River on the valley floor
(610 m a.s.l.). The channel stretching from the Illhorn mountain to the Rhone River has a length of ~6.5 km. On the lowest 4.8
km of the channel 29 check dams are present – this reach has an unconsolidated bed (Schürch et al., 2011; De Haas et al., In
Review). For the last 2 km downstream the channel traverses a large alluvial fan. At the downstream end of the channel an
automated observation station is operated by the Swiss Federal Institute for Forest Snow and Landscape Research (WSL),
which records a myriad of flow properties and collects imagery (McArdell et al., 2007; Schlunegger et al., 2009).

## 3 Methods

### 3.1 Data acquisition

UAV imagery of the study reach was captured during nine surveys between 8 November 2018 and 30 August 2019, covering
the activity of nine debris flows (Table 1). Between each survey one debris flow typically occurred, although for two surveys
the cumulative effect of two debris flows which occurred within a day of each other was captured. The surveys were flown
with a DJI Mavic 2 Pro using Litchi flight planning software. The Mavic 2 Pro is equipped with a 1" CMOS sensor collecting
imagery with a resolution of 20 megapixels. Imagery was captured in two separate flights during each survey, at an altitude of
100 m above the ground surface resulting in a ground sampling distance of 2.5 cm. Flight planning was optimized over time
and differed between surveys. In general, images were captured with a side overlap of 80% and a forward overlap of 70%.
During some surveys nadir images were captured from both channel banks and the middle of the channel and images with a
25° off-nadir camera pitch over the middle of the channel looking in an upstream direction, while during other surveys nadir
and 25° off-nadir images were captured over both the banks and channel. As such, the number of aerial images in the surveys
varied between 168 and 396 (Table 1). A total of 29 anthropogenic and natural terrain features were used as ground control
points (GCPs), including manholes, road surface marks, cobbles and boulders (Fig. 1). These GCPs were measured with a
Leica network RTK GNSS system with ~2 cm accuracy.



### 3.2 Data processing

The SfM processing was performed using Agisoft Metashape Pro (v. 1.5.2). Our general procedure in Agisoft Metashape Pro

was as follows. Photo alignment was performed at high-quality settings using 60,000 and 20,000 key and tie point limits, respectively. The alignment was typically optimized by removing (1) tie points present in less than three images, (2) with a reconstruction uncertainty larger than 50, (3) with a projection accuracy larger than 10, and (4) a reprojection error larger than 1. After each of these steps the alignment was optimized using adaptive camera model fitting. GCPs were added between filter steps 3 and 4. Dense clouds were generated at high quality and mild depth filtering. Orthophotos were generated using a

smoothed mesh of the sparse point clouds, and exported with a ground sampling distance of 2.5 cm. To filter erroneous points and overhanging vegetation from the dense point clouds we adopted the approach of De Haas et al. (In Review) using LAStools (rapidlasso GmbH). The procedure removes low noise and filters overhanging vegetation, while retaining natural detail in the channel, and mostly avoids clipping at steep sections at the channel banks and check dams. Low noise points are typical for dense point clouds generated using UAV photogrammetry, and were filtered by removing points more than 0.1 m below a

smoothed 20th height percentile surface with a step size of 0.5 m. Overhanging vegetation was removed by classifying ground points using the lasground functionality in LAStools with 'ultra fine' settings. This setting only effectively removes overhanging and sparse vegetation in the channel, but retains most of the fine details in the channel at the expense of including dense vegetation in geomorphologically inactive areas which were not of interest to our analysis. Filtered points were rasterized into a DSM with a ground sampling distance of 5 cm.

We compared three workflows for generating DSMs and change detection (Fig. 2): (1) the classical approach where each time step was individually processed using GCPs; (2) the co-alignment approach where the imagery from the nine surveys is processed together without GCPs (cf. Cook and Dietze, 2019); and (3) the co-alignment approach with GCPs. The co-alignment approach for processing UAV imagery for optimized changed detection was first proposed by Cook and Dietze (2019), but where Cook and Dietze applied this method to two time steps and without using GCPs – we take the co-alignment

approach a step further and apply it to nine time steps while also using GCPs. Following Cook and Dietze we imported the photographs from the nine surveys into a single chunk in Metashape Pro and performed the point detection and matching, initial bundle adjustment, and optimization steps on the combined set of photographs. Following the alignment and optimization steps, the photos from the different surveys were separated by creating nine duplicates of the original chunk and keeping only those photos from the relevant time step calculating dense clouds for each survey while preserving common

position information and camera calibrations.

### 3.3 Data analysis

To analyse the accuracy of change detection using the three approaches, we selected 48 validation points that consisted of anthropogenic and natural terrain features that were unchanged during the study period (Fig. 1). These terrain features include cobbles, boulders, manholes, and road surface marks. To quantify the accuracy of the models we then quantified the offset,



both in the xy and z directions, between these points on the DSMs of the nine time steps. This quantification was done by calculating the mean offset in xy and z direction, relative to the mean location of the points in the nine surveys. We further evaluated offset as a function of the distance to the most nearby GCP, to assess if and how the accuracy of the change detection varies with distance from the most nearby GCP for the two approaches that include GCPs. We did not compare the point clouds directly because the channel changed substantially between events, and the banks were largely covered by forest with an

irregular surface not suitable for accurate matching of the point clouds of the different surveys.

## 4 Results

Agisoft Metashape Pro managed to co-align the nine surveys without any problems, despite the substantial changes in the debris-flow torrent during the survey period (De Haas et al., In Review) and the large area of forest in our study area that partly changed in appearance between the autumn, spring, and summer flights. We did find that surveys 6 and 8 were relatively

poorly aligned, both when processed individually and when co-aligned. These surveys had relatively large reprojection errors and a relatively low number of tie points (Table 1). Surveys 6 and 8 had relatively large mean reprojection errors in comparison to the mean reprojection error in the other surveys. In addition, surveys 6 and 8 had ~65,000 and 170,000 tie points, respectively, compared to an average of 360,000 tie points in the other surveys.

The classical approach wherein surveys are processed individually using GCPs results in the largest offset between the DSMs

of the nine surveys in both the xy and z directions (Fig. 3a,b). The mean offset between the surveys in the xy direction for the classical approach is 0.17 m and for both the co-alignment with and without GCPs the offset is 0.09 m. In the z direction the mean offset between the surveys is 0.25 m for the classical approach, while it is 0.08 m for the co-alignment without GCPs and 0.06 m for the co-alignment with GCPs. As such, co-aligning the surveys improves the xy offset by approximately a factor 2, while the improvement in the z direction approaches a factor 3 for co-alignment without GCPs and 4 for co-alignment with

GCPs.

For both the classical approach and co-alignment with GCPs, the offset in the xy and z directions increases with distance from the nearest GCP. This trend is more pronounced for the z offset compared to the xy offset. The distance of the validation points to the nearest GCP was relatively small in our survey, ranging from a few m to almost 100 m, and we find that the offset most strongly increases when the distance to the nearest GCP exceeds 10 m, typically by a factor three moving from a distance of

~10 to ~100 m.

If we break down the errors per survey we see that surveys 6 and 8 have large offsets in both the xy and z directions for the models generated via the classical approach. The xy offset of 0.4 m for survey 6 is approximately a factor 4 larger than the typical offset of ~0.1 m. The z offset for survey 6 is even more dramatic with a value of 0.97 m compared to a typical value of 0.15 m for the models made via the classical approach. For survey 8 the offset in the xy and z directions exceeds the typical

offset of the other surveys by approximately a factor 2. Interestingly, when applying the co-alignment approach, either with or without GCPs, the xy and z offset of these surveys is strongly reduced. While the offset of surveys 6 and 8 is still slightly





larger than that of the other surveys in this case, it has dropped to an acceptable offset of 0.1 m in both the xy and z directions. This shows that co-alignment forces otherwise poorly aligned surveys onto the more accurate common geometry set by the other surveys, thereby strongly decreasing the offset between the surveys and increasing the accuracy of topographic change

detection.

This effect is further projected in the calculated volume changes in the study reach as a result of the debris-flow activity. Here, we see a fairly consistent trend between the co-alignment approaches with and without GCPs (Fig. 5). However, the poorly-aligned models of surveys 6 and 8 processed with the classical approach, lead to large errors in the calculated volume change (Fig. 5). The reason behind these mis-calculations is illustrated in Figure 6, which illustrates how the classical approach leads

to a systematic bias in the topographic change of 2 m over a part of the study reach. Importantly, in topographic change analyses poorly-aligned models affect both the comparison with the before and after survey, thereby propagating errors in two directions rendering a large portion of a dataset unreliable even when only a small portion of the surveys is of relatively low quality.

## 5 Discussion and conclusions

Our results show that co-aligning multiple surveys through UAV-SfM leads to more accurate topographic change detection compared to the classical approach where each survey is processed individually. Using GCPs in the co-alignment procedure does lead to a slightly increased topographic change detection accuracy compared to co-aligning without GCPs, but both approaches perform well with an offset in the xy and z directions below 0.1 m. Such accuracy is more than sufficient for most applications of change detection in a wide range of fields including geosciences, forestry, ecology, archaeology, mining and

engineering (e.g., Lucier et al., 2014; Torres-Sánchez et al., 2014; Braun et al., 2015; Lee and Choi, 2015; Qin et al., 2016; James et al., 2017; Hemmelder et al., 2018). Co-alignment fits multiple surveys onto a common geometry so that any distortions become consistent between the co-aligned surveys. Potential errors do therefore not influence comparisons between the 3D models of the surveys, such that their comparative accuracy is much higher and topographic change detection is more accurate. The increase in comparative accuracy is due to the generation of common tie points between photographs from

different surveys, enforcing a common geometry between the different surveys (Cook and Dietze, 2019). Our results show that especially when multiple surveys are co-aligned, this forces poorly aligned surveys, which retain this poor quality even after adding a large number of GCPs, onto the more accurate common geometry set by the other surveys (Fig. 4). This strongly improves the accuracy of topographic change detection for those surveys that are poorly aligned using the classical approach. A key limitation of the co-alignment method is that in order to get a successful alignment, tie points linking the photos from

different surveys must be detected and false matches must be avoided (Cook and Dietze, 2019). However, sufficient tie points may not be generated when an area changes too much in appearance between surveys or when there have been too much changes in the area of interest. Yet, for our study area co-alignment was successful, despite that substantial changes occurred in the torrent bed (De Haas et al., In Review), and changes in appearance from autumn to summer in the forest that covers the



largest part of our study area. This suggests that co-alignment is a fairly robust method that can be successfully employed even
in areas where only restricted areas are stable over time.

While Cook and Dietze (2019) suggested that co-alignment without GCPs can be used for change detection with a level of detection comparable to that of a survey grade GCP-constrained pair of models, for our dataset combining nine surveys, co-alignment without GCPs outperforms the classical approach where surveys are processed individually with GCPs. This can be attributed to our finding that the co-alignment procedure forces surveys with poor alignment into a common geometric
framework, strongly limiting their offset from the other surveys. Still, as pointed out by Cook and Dietze (2019) one has to bear in mind that 3D models created through co-alignment in the absence of GCPs may still contain absolute errors and distortions such as doming (James and Robson, 2014; Carbonneau and Dietrich, 2017). Moreover, the absolute location of models created in the absence of GCPs typically has a low accuracy, inhibiting comparison with external sources such as lidar topography (e.g., Neugirg et al., 2016; Izumida et al., 2017).

In conclusion, given the much higher accuracy of topographic change detection obtained by co-alignment approaches, especially when combined with GCPs, compared to the classical approach where surveys are processed individually with GCPs, we advocate that co-alignment should become the new norm in UAV-SfM based topographic change detection. The co-alignment approach was found to enhance the accuracy of topographic change detection by a factor 3-4 in the z direction and a factor 2 in the xy direction. A particularly large advantage of co-alignment is that it forces poor quality (parts of) surveys
to the more robust common geometry set by the other surveys, which strongly increases the comparability of the surveys and the accuracy of topographic change detection. The co-alignment approach can be applied and be fully automated in most SfM software packages, such that there should be no technical limitations in applying co-alignment instead of the classical approach processing each survey separately providing sufficient processing power.

## 6 Acknowledgements

This work was supported by the Netherlands Organisation for Scientific Research (NWO) (grant 016.Veni.192.001 to TdH).





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



**Table 1: Survey characteristics and dates of captured debris flows.**

| Survey | Acquisition | Debris flow(s) | Photos (aligned/total) | | | Tie points | | | Mean projection error (pix) | | |
|---|---|---|---|---|---|---|---|---|---|---|---|
| | | | GCP | CA | CA+GCP | GCP | CA | CA+GCP | GCP | CA | CA+GCP |
| 1 | 11 Nov 2018 | - | 187/209 | 183/209 | 183/209 | 373,613 | 395,098 | 392,593 | 0.54 | 0.50 | 0.75 |
| 2 | 30 April 2019 | 4 Dec 2018 | 396/396 | 245/396 | 245/396 | 508,452 | 320,344 | 324,734 | 0.72 | 0.56 | 0.62 |
| 3 | 16 June 2019 | 10 June 2019 (2) | 292/292 | 226/292 | 226/292 | 342,393 | 308,593 | 305,060 | 1.45 | 0.51 | 0.54 |
| 4 | 22 June 2019 | 21 June 2019 | 320/320 | 306/320 | 306/320 | 388,484 | 433,148 | 444,214 | 0.47 | 0.48 | 0.51 |
| 5 | 4 July 2019 | 2 July 2019 & | 240/331 | 228/292 | 228/292 | 393,895 | 396,461 | 396,260 | 0.47 | 0.49 | 0.52 |
| | | 3 July 2019 | | | | | | | | | |
| 6 | 17 July 2019 | 15 July 2019 | 121/207 | 112/207 | 112/207 | 56,380 | 68,552 | 68,478 | 1.86 | 0.63 | 1.21 |
| 7 | 30 July 2019 | 26 July 2019 | 204/204 | 170/204 | 170/204 | 544,434 | 250,151 | 250,220 | 0.87 | 0.47 | 0.55 |
| 8 | 14 Aug 2019 | 11 Aug 2019 | 151/168 | 142/168 | 142/168 | 249,589 | 138,849 | 141,354 | 0.54 | 0.64 | 1.25 |
| 9 | 30 Aug 2019 | 20 Aug 2019 | 168/168 | 148/168 | 148/168 | 357,270 | 261,060 | 268,837 | 0.55 | 0.49 | 0.55 |






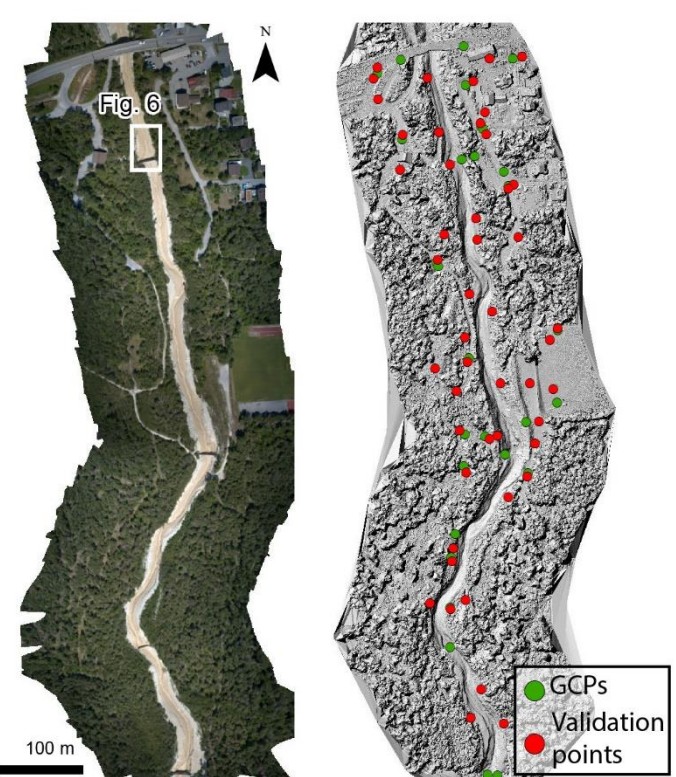

**Figure 1: Orthophoto and hillshaded DSM of the study area, showing the locations of the GCPs and validation points. Imagery from survey 7 (30-07-2019). The centre of the study area is located at 46°18'25.55''N and 7°38'2.97''E.**





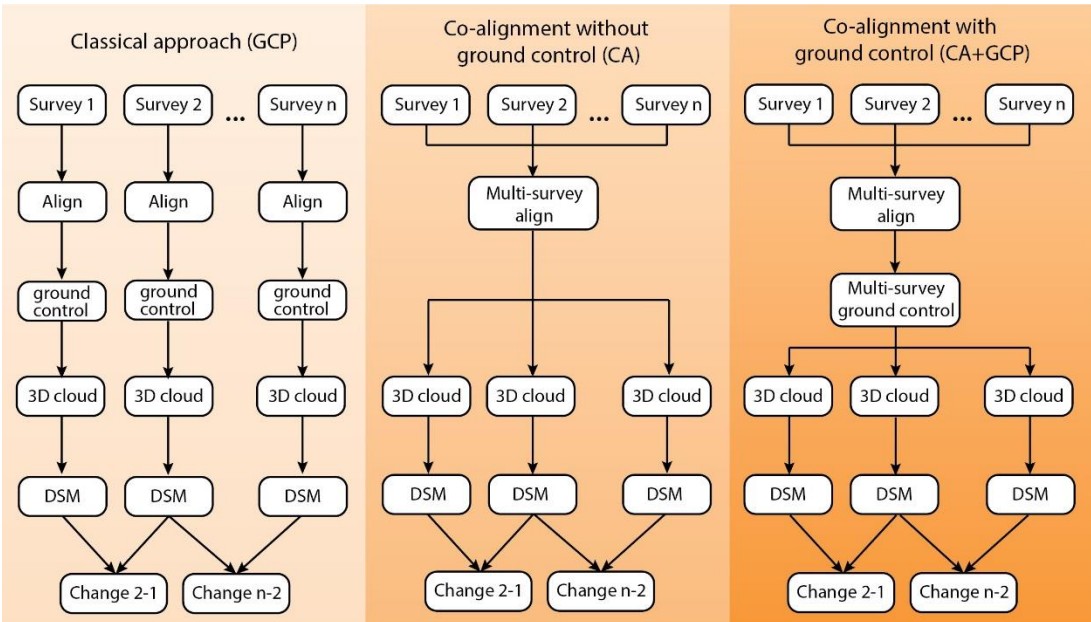

**Figure 2: Generic workflows of the three tested approaches for UAV-SfM based topographic changes detection: (1) the classical approach where each survey is processed individually and ground control is applied (GCP); (2) co-alignment without ground control (CA); (3) co-alignment with ground control (CA+GCP).**





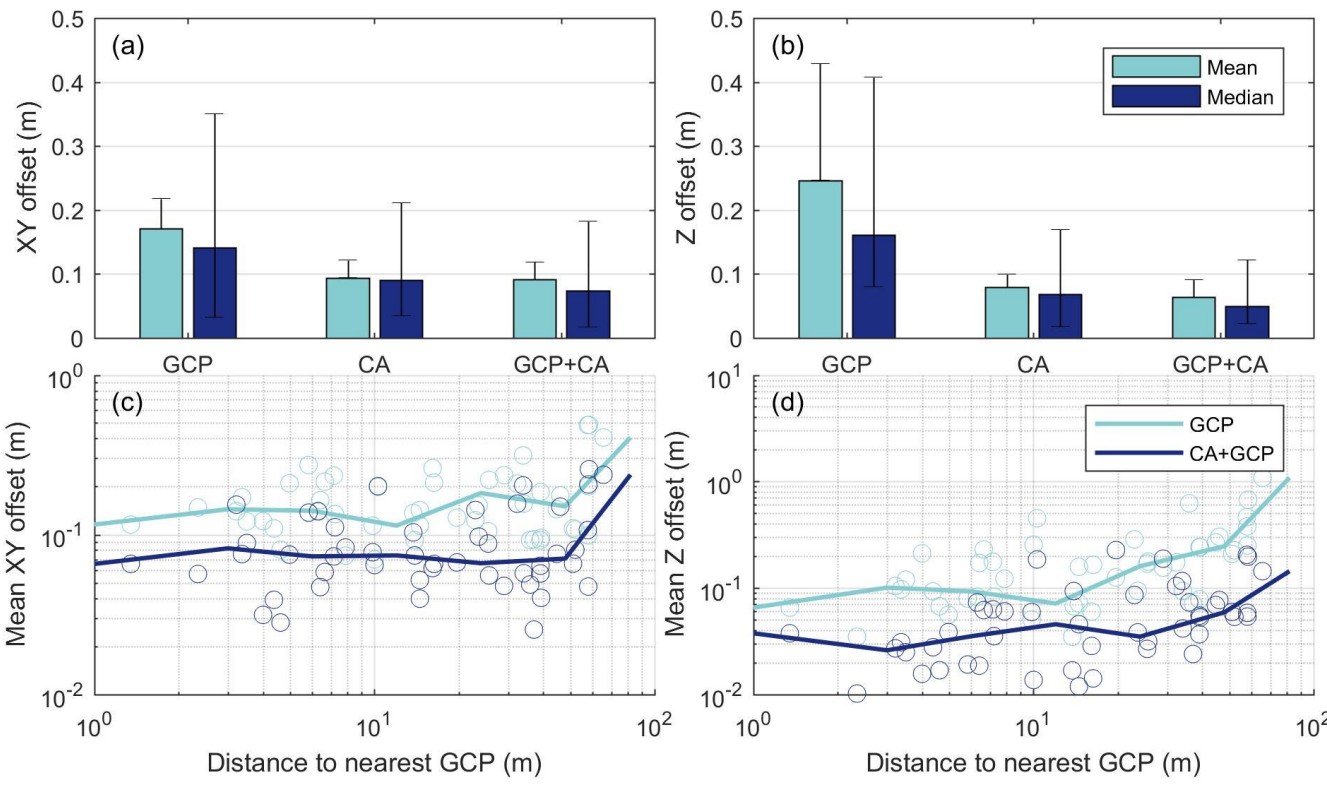

**Figure 3: Mean and median offset in the (a) xy direction and (b) z direction for all surveys combined, processed with the classical approach where surveys are processed individually using GCPs (GCP), co-alignment without GCPs (CA) and co-alignment with GCPs (CA+GCP), showing a higher accuracy of the co-alignment approaches. (c,d) Distance to the nearest GCP versus the mean offsets in the xy and z directions, respectively, showing that the 3D model accuracy decreases with distance to the nearest GCP.**





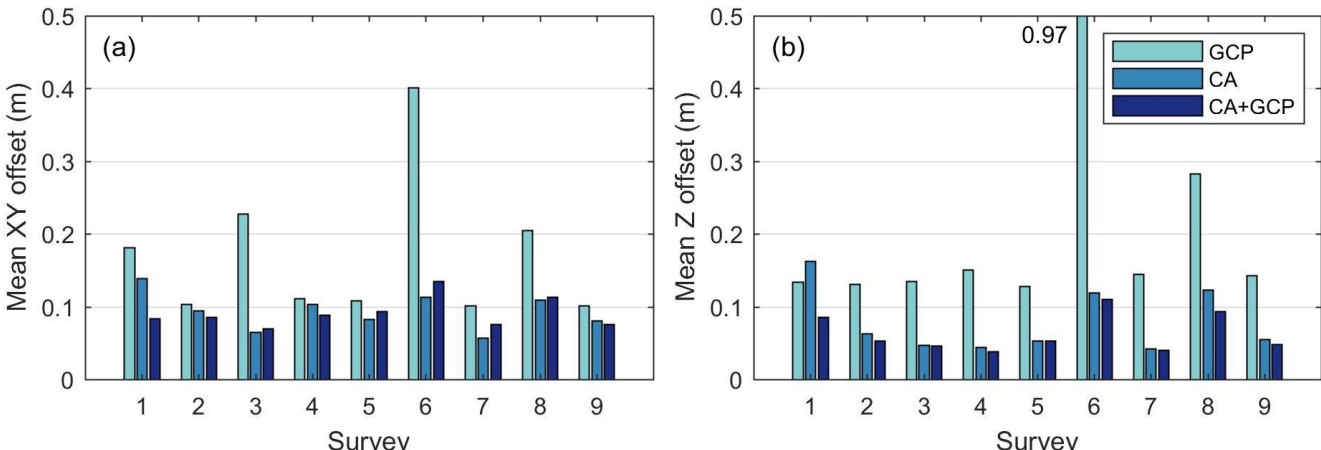

**Figure 4: Mean offset in the (a) xy direction and (b) z direction for the nine surveys processed with the classical approach where surveys are processed individually using GCPs (GCP), co-alignment without GCPs (CA) and co-alignment with GCPs (CA+GCP). The image shows that the co-alignment approaches lead to a lower offset in the xy and z directions compared to the classical approach, especially for the poorly aligned surveys 8 and 9.**



Earth **Surface**
**Dynamics**
Discussions



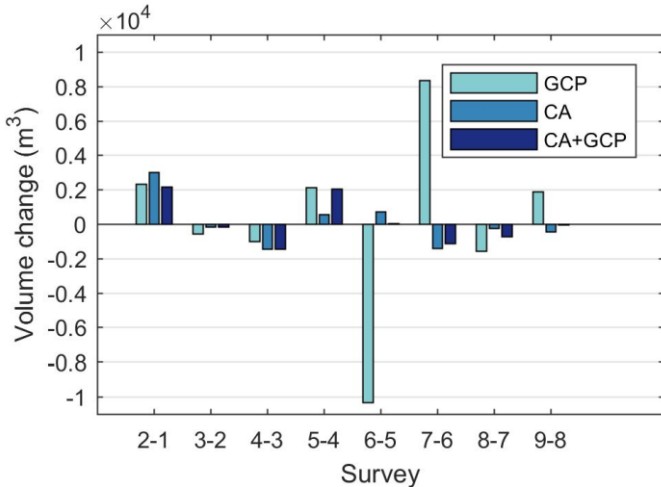

**Figure 5: Volume change in the studied reach of the Illgraben torrent, as calculated via the classical approach where surveys are processed individually using GCPs (GCP), co-alignment without GCPs (CA) and co-alignment with GCPs (CA+GCP). The image highlights how the low-quality surveys 6 and 8, when processed with the classical approach, negatively affect the volume change**
320 **calculations leading to erroneous values of volume change. The image further highlights that in geomorphic change detection errors as a result of a poorly-aligned survey are propagated in two directions, as the poorly-aligned survey is compared to the before and after survey.**





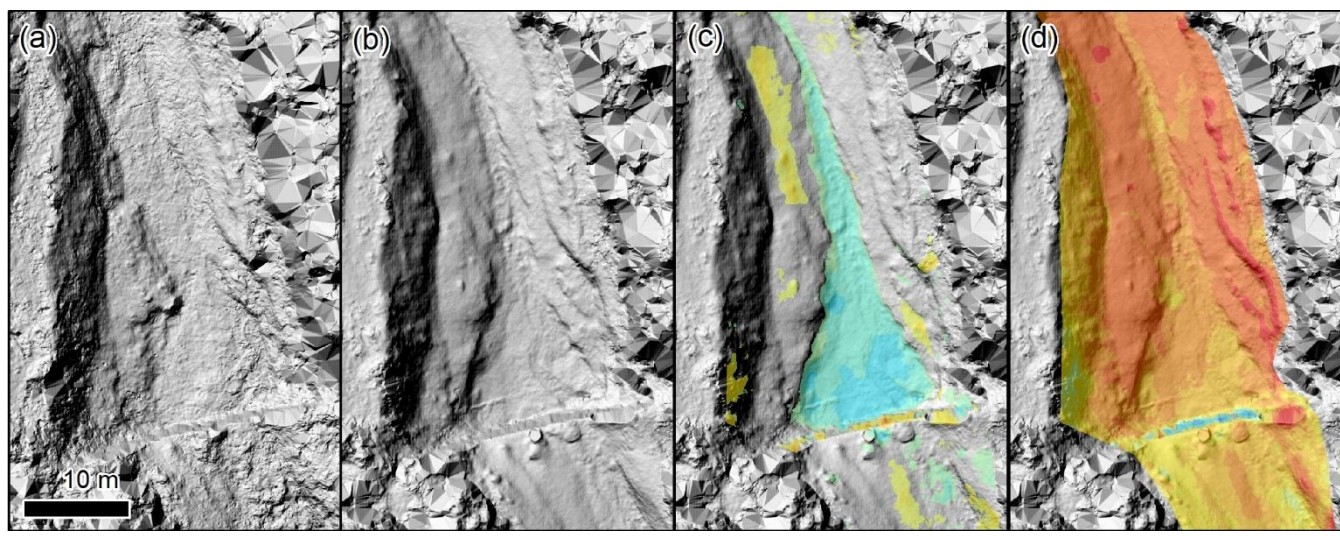

Figure 6: Topographic change detection between surveys 6 (17-07-2019) and 7 (30-07-2019), as calculated via co-alignment with GCPs and with the classical approach processing each survey individually using GCPs. (a) Hillshaded DSM of survey 6 – note the noise in the image. (b) Hillshaded DSM of survey 7. (c) Topographic change calculated via co-alignment with GCPs. (d) Topographic change calculated with the classical approach processing each survey individually using GCPs. Warm colours denote deposition and cold colours denote erosion, ranging from -4 m (dark red) to 4 m (dark blue) channel-bed elevation change, values ranging between -0.25 and 0.25 m are transparent. See Figure 1 for location. Orientation of all images is to the north, and flow is from bottom to top. Note how the channel change in panel d is uniformly overestimated by ~2 m, as a result of the poor quality of the 3D model of survey 6 calculated via the classical approach.