# Peer review of "Short Communication: Optimizing UAV-SfM based topographic change detection with survey co-alignment"

_Earth Surface Dynamics, 2020_

## Referee Comment (RC1) · Anonymous Referee #1 · 18 Jul 2020

The manuscript discusses the advancement of the already introduced method of time-SIFT/ co-alignment to improve multi-temporal changed detection with SfM. The authors investigate the differences in performance when using the common approach of change detection (reconstructing and referencing each dataset separately), a co-alignment approach with more than two datasets and an approach that co-aligns and implements GCPs. Their results suggest a performance improvement of the latter two methods. The manuscript is well structured and written in a manner that makes the content easy to understand.

I do like the idea that this study eventually implements GCPs, which was thus far miss-

ing in the previous studies introducing time-SIFT. This consideration would be needed to highlight that the usage of co-alignment should be preferred over the standard workflow (i.e. separately processing each dataset) as long as sufficient stable areas are visible in the area of interest. However, to indeed evaluate how the usage of GCPs in time-SIFT might outperform the common approach the comparison with an independent method is missing. For instance, the usage of check points (which are GCPs that are not implemented in the bundle adjustment to reconstruct the 3D geometry) or TLS datasets would be a possibility to check whether the final models processed with time-SIFT and GCPs reveal similar or better absolute accuracies compared to the common approach. The authors have distributed quite a few GCPs and could therefore consider using some off them as check points.

The authors do compare their methods using the average of all models of stable points in the area of interest. This is very sufficient to investigate whether relative changes are detected correctly. However, it is not possible to assess absolute accuracies of change detection, as the scale can still be wrong. By implementing GCPs an improvement of absolute accuracies and thus potential scaling errors should be mitigated, which can remain when co-alignment is used without geo-referenced markers and solely relying on the camera positions of poor quality (if not RTK or PPK capable UAVs are considered). Without geo-referencing (directly or indirectly) absolute magnitudes of change still might be false. In the current manuscript version, the authors solely see an improvement regarding the relative accuracy by using GCPs as they do not compare it to independent data, but the actual potential improvement regarding scale/absolute orientation accuracies is yet to be shown. And the improvement of relative accuracies has already been demonstrated by Cook & Dietze (2019) and Feurer & Vinatier (2018). Furthermore, the illustrated improvement of relative change detection when using GCPs in time-SIFT does not seem to be statistically proven (when looking at the average change and its standard deviation in figure 3, where the latter is larger than the former). Did the authors check whether the difference between approach 2 and 3 is significant?

The authors further highlight that they use not just two datasets but in total nine, which allows to also improve surveys with weak image geometries. This is an important note. The authors nicely illustrate that the higher the number observations (i.e. number of tie points across more images) the better the reconstruction (which is well-known to photogrammetry). However, I think the manuscript would benefit if the authors might also have a closer look at that finding by gradually increasing the number of surveys to check (for their case study) if there might be a point after which further survey implementations do not improve the accuracies anymore. Please, see some more specific comments below.

Minor comments:

18-20 But only for scenarios with enough stable areas?

34-35 Maybe refer to direct and indirect referencing, respectively.

36-38 Besides georeferencing accuracy also the image geometry is important for a successful reconstruction.

41-42 Please, also refer to time-SIFT as it describes the same approach and was introduced first.

46-47 The mentioned study could not describe how well it performs absolutely because the showcase had no absolute references. But the relative error assessment was possible. – I would use the terms absolute and relative accuracy here.

51 The authors claim to assess the influence of a larger number of surveys. But I think, they need to show this aspect in a more detailed analysis, e.g. considering the gradual increase of surveys.

58 Is the workflow indeed fully automatic. E.g. how did the authors split the model after alignment (via the python API?) or how did they provide the GCPs (did they use automatic detection of coded markers?)?

90-91 Why did the authors not use photogrammetric markers (especially considering that they went around the area of interest anyway to measure points) which are known to improve the reconstruction accuracy significantly due to subpixel measurement capabilities in image?

96-97 Please, explain how these error metrics are computed.

98 What is the advantage of adaptive camera model fitting? What is the difference to the standard approach?

89-99 Why are GCPs added between filter step 3 and 4 and not just after filtering?

100 Why is the orthophoto with sparse points used instead of dense? You can expect quite a lot of artefacts because the true 3D geometry is smoothed too strongly/not captured with enough resolution with the sparse point cloud.

103 What are the low noise points?

106 How does the lasground tool work?

109 How was the point cloud rasterized (e.g. using IDW)?

115 What is expected by increasing the number of surveys? The explanation of the reason for investigation is missing here. Do the authors expect an improvement due to even more potential tie points across more images?

110-120 Are the GCPs measured in all images across time or only in one survey? Please, also assess the latter case because this could be a significant improvement regarding the time and potential availability of GCPs (e.g. maybe it is not possible to set them up each time).

126 Thus, you evaluated the precision (if you compare the distances to the mean of the entire dataset) rather than the accuracy.

126-128 Please, also refer to the already existing literature discussing the error dependence to the distance to the GCPs. For instance, see Tonkin & Midgley (2016).

128-129 I am not sure if this sentence is correctly stated because you do compare the point clouds directly using the mean of all nine point clouds at your selected points. Might you mean that you do not compare entire regions rather than just single points?

135-137 What is the difference between reprojection error and mean reprojection error? If the authors refer to the value provided by MetaShape it would be root means squared reprojection error.

146-147 This finding is not new and well known in conventional photogrammetry as well as more recent work. Please, refer to these studies (e.g. Tonkin & Midgley (2016) and/or standard work (e.g. Kraus, 2007)).

165-169 I do not understand this statement. How does a small (thus local) portion of the survey influence the entire/global change detection?

170-171 As shown by Cook & Dietze (2019) and Feurer & Vinatier (2018)

171-173 However, the comparison to independent references is still missing, which might indicate the more important aspect if the integration of GCPs does improve the absolute accuracy as well as the scaling of the combined model, which would be important for the volumetric change calculation.

176-177 I might rephrase this sentence stating that the used co-alignment allows for more reliable change detection because during the reconstruction all images (from all surveys) are optimized within the same adjustment, using homologous image points covering several time-steps and therefore resulting in a joint camera geometry. Potential systematic errors are therefore spatially consistent.

References:

Tonkin, T., Midgley, N. (2016). Ground-Control Networks for Image Based Surface Reconstruction: An Investigation of Optimum Survey Designs Using UAV Derived Imagery

and Structure-from-Motion Photogrammetry. Remote Sensing

Kraus, K. (2007). Photogrammetry: Geometry from Images and Laser Scans, 2nd edition, De Gruyter, Berlin, Germany

---

## Referee Comment (RC2) · Benjamin Purinton (Referee) · 10 Aug 2020

General Comments:

The authors advocate for a modified method of change detection using UAV-SfM time series. Whereas the traditional approach uses each survey individually aligned to GCPs, the authors build upon and expand the work of Cook and Dietze (2019) in this journal to suggest a combination of GCPs and co-alignment over many (more than two) time steps. This manuscript is well written and well placed within the context of recent UAV-SfM studies. Despite being a short communication, the manuscript would benefit from some additional details, clarifications, and modifications that should all fall

in the realm of minor revisions prior to acceptance.

I understand that the authors lack a control dataset to test absolute accuracies, but I tend to agree with the other reviewer that some GCPs could be used as check points on absolute accuracy between the three approaches, listing (or better tabulating) a simple metric like mean and standard deviation ought to suffice. Furthermore, the issue of GCPs is of great concern especially when surveying large areas where there may be difficulty in finding and reaching stable, clearly visible points to lay out targets and measure using dGPS. I am particularly interested in how increasing the number of GCPs (and their spatial arrangement in the area) affects the quality of CA+GCP method. Figure 3c,d go in this direction with plots showing increasing error with distance from GCPs. Interestingly the z error has the clearest trend with distance from GCP, whereas the xy error trend is less clear and only shows a decrease in accuracy for a few points at distances > ∼50 m. Would these trends be better presented in non-log space? It looks like a lot of the accuracies are sub-cm, so they may be "insignificant" (or at least negligible) for most geomorphic change detection studies, thus a linear (or semilog-x) plot would highlight the larger (1 cm - 1 m) inaccuracies that are of greater concern. If a log axis is preferred then the text should at least highlight and state the lower importance of these sub-cm inaccuracies. The xy trend may also be more apparent if less GCPs were used and the x-axis of that plot extended to some higher values (i.e., greater distances). If similar accuracy can be achieved with GCPs placed 80 m apart versus 20 m (e.g., accuracy ∼1 cm) then that's a significant improvement in field work time.

If the authors aim for reproducibility and standardization of UAV-SfM change detection then some additional details of the steps taken should be included. I have a couple suggestions here, which I mostly highlight in the line changes below. One point here: in the Section 3.2 Data processing, I suggest a nested alpha-numeric list (e.g., (1), (a), (b),...(2), (a), (b),...) rather than the steps in paragraph form. This will read more like a manual of the steps to follow. If these and previous suggestions are increasing

the length significantly, I recommend removing Figure 2. I think the difference between these methods is quite clear in the text and the flow-chart is not necessary. Alternatively, the current Figure 2 could be replaced by a flow-chart of only the proposed method similar to Figure 2 in Cooke and Dietze (2019). This may negate the need for the alpha-numeric list, but that is for the authors to see and decide.

One missed study from this journal that adds some nice context is Duro et al. (2018). They used the traditional GCP approach without co-alignment on 8 surveys. It would be good to include this reference somewhere in the introduction or in the discussion as an example of the previously used (GCP only) technique for UAV-SfM surveys over many time steps (rather than just two). Table 2 and Figure 6 in that study could also provide some discussion comparisons with the absolute accuracies found using check points in the present study.

A last general comment: Agisoft version 1.6 (in the present study version 1.5.2 is used) added a point confidence based on the number of depth maps a point appears in. I suggest the authors highlight this new feature to call attention to it. For instance this new feature could be used in further filtering of the point clouds to only extract high confidence points and/or as weights in interpolation schemes. There is no need to do any re-processing using the newer version, but this is certainly an important new feature that ought to be explored in future research. This could just be a few sentences in the discussion.

Line Changes:

L33: "differentiated" to "differenced"

L45: should "reduces the accuracy" instead say "increases the accuracy"?

L46-47: I think the terms relative and absolute accuracy (as opposed to comparative and external) are more common. It is fine to continue using comparative and external but maybe quickly define, e.g., "comparative – or relative – accuracy... external – or

absolute – accuracy"

L57: remove hyphen from "common-practice"

L94: Please include the specs of the computer used (e.g., RAM, number of CPUs, presence or absence of a GPU) and approximate processing times (this could come at the end of the paragraph). I'm especially curious about how long the 9 survey block co-alignment took on whatever computer was used.

L95: Were these steps done using the Agisoft Python API or strictly through the Agisoft GUI? If Python, consider including a GitHub repository with the code. Or at least a statement saying whether the API or GUI was used. The iterative steps of tie point removal and re-alignment would be especially ideal for a Python script. As earth surface processes research moves towards increasing reproducibility the move to scripting as opposed to clicking steps is vital, or at least more manual-like details (e.g., the alphanumeric list or detailed flow-chart I suggest in general comment).

L98-99: All GCPs were added between steps 3 and 4? Or just a few and then kept adding more with each iteration? This could be the point where 10, 25, 50, 75, 90% (or so) of the GCPs are added each time to see the improvement / changes in accuracy.

L101 to end of paragraph: include a few more details on the lastools commands used (e.g., lasground, what were all the parameters?), and also include details of the "rasterization" scheme. Here "rasterized" should be changed to "interpolated" and the method of interpolation should be stated (e.g., IDW?).

L103: Should "low noise" say "high noise"? Or maybe just "low accuracy".

L113: "changed" should be "change"

Figure 3: What are the error bars in a and b? I suggest removing the median bars and just use the mean, since the median is never referenced in the text and Figure 4 uses the mean. However, the median error bars show large overlap, which is concerning and would seem to limit the interpretation of significant improvements. Do the authors
have a response to this concern?

Figure 3: In the caption "accuracy decreases with distance" should perhaps be "accuracy decreases with increasing distance"?

L147: In the xy direction this decrease in accuracy is only apparent in the few data-points > ~50 m from a GCP (if one considers < ~1 cm inaccuracies as negligible in this context). Be more specific about these distance breaks in the text, or perhaps use the suggested gradual addition of GCPs to highlight this. Also, what are the fitted lines in Figure 3c,d? Is this a moving average? Please state this.

L146-160: In the results please reference Figures 3 and 4 where appropriate.

L163: The "large errors" in volume change don't refer to any reference data. I'm not sure it's appropriate to refer to the volume change errors without a control (e.g., lidar). Instead these are relatively (compared with the other alignment techniques) large changes in volume that are outside of expectations. Based on Figure 6, it seems that this error is a visual assessment, which is fine but should be clarified.

Figure 6: It would be helpful to just put a colorbar on this and remove the caption description (+/-4 m in color and +/-0.25 m transparency).

L198: "has" should be "have"

References:

Duró, G., Crosato, A., Kleinhans, M. G., and Uijttewaal, W. S. J.: Bank erosion processes measured with UAV-SfM along complex banklines of a straight mid-sized river reach, Earth Surf. Dynam., 6, 933–953, https://doi.org/10.5194/esurf-6-933-2018, 2018.

---

## Referee Comment (RC3) · Anonymous Referee #3 · 11 Aug 2020

While is gratifying that the workflow we discussed in Cook and Dietze, 2019 (and was proposed by Feurer and Vinatier, 2018) was shown to be effective even when GCPs are used, I found this paper very thin for a stand-alone paper, even a short communication – it feels like something carved out as a least publishable unit when it could be included in a paper that discusses the observed changes (I notice that the authors appear to have such a manuscript in review). While the additional testing of the method is nice, it's not clear that this manuscript adds much new knowledge or methodological development. This is perhaps best exemplified by the discussion, much of which (lines 176-187, 191-199, 204-208) restates points that were already made in Cook and Dietze.

[Figure]

The authors state one the key unanswered questions they address is "how well co-alignment does on a larger number of surveys" and that Cook and Dietze tested only two surveys. This is not the case; while most of the results in Cook and Dietze were from survey pairs, we also applied the method to a set of 4 surveys (shown in Fig. 6 in our paper) and note that including additional surveys could improve results in some cases. While 9 is greater than 4, it's not clear that this is enough of a difference to be noteworthy. This leaves the addition of accurate GCPs as the only new thing (note that Feurer and Vinatier also used GCPS, just not precise ones).

Overall, I am doubtful that this manuscript makes a significant scientific contribution, again best illustrated by the lack of original ideas in the discussion.

---

## Editor Comment (EC1) · Wolfgang Schwanghart (Editor) · 11 Aug 2020

I like to thank all reviewers for their constructive comments on the manuscript by de Haas et al, and I am looking forward to a final response by the authors. In addition, I noticed that the authors referenced own work that is currently in review (De Haas,T., Nijland, W., De Jong, S., M., McArdell, B. W.: How memory effects, check dams, and channel geometry control erosion and deposition by debris flows.Scientific Reports, In Review). Could the authors please supply a link to the paper if now published or consider posting a pre-print?

[Figure]

2020.

---

## Author Comment (AC1) · 2 Oct 2020

(R1.1) The manuscript discusses the advancement of the already introduced method of time-SIFT/ co-alignment to improve multi-temporal changed detection with SfM. The authors investigate the differences in performance when using the common approach of change detection (reconstructing and referencing each dataset separately), a coalignment approach with more than two datasets and an approach that co-aligns and implements GCPs. Their results suggest a performance improvement of the latter two methods. The manuscript is well structured and written in a manner that makes

the content easy to understand.

Reply: We are grateful for the positive evaluation of our manuscript. We thank the reviewer for his/her constructive comments, to which we respond on a point-by-point basis below.

(R1.2) I do like the idea that this study eventually implements GCPs, which was thus far missing in the previous studies introducing time-SIFT. This consideration would be needed to highlight that the usage of co-alignment should be preferred over the standard workflow (i.e. separately processing each dataset) as long as sufficient stable areas are visible in the area of interest. However, to indeed evaluate how the usage of GCPs in time-SIFT might outperform the common approach the comparison with an independent method is missing. For instance, the usage of check points (which are GCPs that are not implemented in the bundle adjustment to reconstruct the 3D geometry) or TLS datasets would be a possibility to check whether the final models processed with time-SIFT and GCPs reveal similar or better absolute accuracies compared to the common approach. The authors have distributed quite a few GCPs and could therefore consider using some off them as check points.

Reply: Thanks for this interesting suggestion. We have now added an analysis of the absolute accuracy by including two extra scenarios, wherein we use 10 GCPs in the SfM and use the remaining 19 GCPs as check points. This analysis shows that co-alignment without GCPs leads to very low absolute accuracies despite having high relative accuracy. When using GCPs, the absolute accuracies in xy and z directions are slightly higher when applying co-alignment than when processing each time step individually. The differences in absolute accuracies between these models are relatively low and statistically marginally significant. See the newly introduced section 4.2 for more details.

(R1.3) The authors do compare their methods using the average of all models of stable points in the area of interest. This is very sufficient to investigate whether relative

changes are detected correctly. However, it is not possible to assess absolute accuracies of change detection, as the scale can still be wrong. By implementing GCPs an improvement of absolute accuracies and thus potential scaling errors should be mitigated, which can remain when co-alignment is used without geo-referenced markers and solely relying on the camera positions of poor quality (if not RTK or PPK capable UAVs are considered). Without geo-referencing (directly or indirectly) absolute magnitudes of change still might be false. In the current manuscript version, the authors solely see an improvement regarding the relative accuracy by using GCPs as they do not compare it to independent data, but the actual potential improvement regarding scale/absolute orientation accuracies is yet to be shown. And the improvement of relative accuracies has already been demonstrated by Cook & Dietze (2019) and Feurer & Vinatier (2018). Furthermore, the illustrated improvement of relative change detection when using GCPs in time-SIFT does not seem to be statistically proven (when looking at the average change and its standard deviation in figure 3, where the latter is larger than the former). Did the authors check whether the difference between approach 2 and 3 is significant?

Reply: As discussed in the comment above we have added an assessment of the absolute accuracy. Moreover, we have now included statistical analysis (t-test) where relevant (e.g., sections 4.1 and 4.2).

(R1.4) The authors further highlight that they use not just two datasets but in total nine, which allows to also improve surveys with weak image geometries. This is an important note. The authors nicely illustrate that the higher the number observations (i.e. number of tie points across more images) the better the reconstruction (which is well-known to photogrammetry). However, I think the manuscript would benefit if the authors might also have a closer look at that finding by gradually increasing the number of surveys to check (for their case study) if there might be a point after which further survey implementations do not improve the accuracies anymore.

Reply: One of the objectives for using co-alignment is to improve the results for surveys with weak image geometries and our analysis shows they benefit most from the the stronger common geometry when applying a co-aligning approach. This finding is to a large degree related to their poor performance when processed independently. While the exact response to the proposed scenario could be interesting (as are studies on trying out co-alignment on many more scenarios including different imaging conditions, sensors, flight plans etc), they are beyond the aim of this manuscript and could dilute the focus. We feel such studies would warrant a paper on its own, as it involves including a lot of scenarios and processing time.

Minor comments: (R1.5) 18-20 But only for scenarios with enough stable areas?

Reply: This is indeed an important restriction to the method, which we now mention explicitly here: "Based on these results we advocate that co-alignment, preferably with GCPs, should become the common-practice in high-accuracy UAV-SfM topographic change detection studies for projects with sufficient stable areas.".

(R1.6) 34-35 Maybe refer to direct and indirect referencing, respectively.

Reply: Done.

(R1.7) 36-38 Besides georeferencing accuracy also the image geometry is important for a successful reconstruction.

Reply: We agree, but do not think it is relevant to mention this here.

(R1.8) 41-42 Please, also refer to time-SIFT as it describes the same approach and was introduced first.

Reply: Done.

(R1.9) 46-47 The mentioned study could not describe how well it performs absolutely because the showcase had no absolute references. But the relative error assessment was possible. – I would use the terms absolute and relative accuracy here.

Reply: Done.

[Figure]

(R1.10) 51 The authors claim to assess the influence of a larger number of surveys. But I think, they need to show this aspect in a more detailed analysis, e.g. considering the gradual increase of surveys.

Reply: See our response to the last main comment above.

(R1.11) 58 Is the workflow indeed fully automatic. E.g. how did the authors split the model after alignment (via the python API?) or how did they provide the GCPs (did they use automatic detection of coded markers?)?

Reply: Indeed, GCPs need to be included by hand. We have therefore replaced "fully automated" with semi-automated in this sentence.

(R1.12) 90-91 Why did the authors not use photogrammetric markers (especially considering that they went around the area of interest anyway to measure points) which are known to improve the reconstruction accuracy significantly due to subpixel measurement capabilities in image?

Reply: This has a practical reason. This study was originally designed to enable rapid response following debris-flow activity in the Illgraben torrent. By using anthropogenic and natural terrain features we did not have to first install and measure GCPs after each debris flow, but could obtain areal imagery for DSM generation with a minimal lag time between debris-flow event and measurement.

(R1.13) 96-97 Please, explain how these error metrics are computed.

Reply: This is standard procedure in Agisoft Metashape (and previously Photoscan). These metrics are generated by the gradual selection tool in the software. Since this is a common procedure in Agisoft Metashape we do not find it necessary to include this explanation in the manuscript. We refer to the Agisoft Metashape manual for further details on the selection procedure.

(R1.14) 98 What is the advantage of adaptive camera model fitting? What is the difference to the standard approach?

Reply: The advantage of adaptive camera model fitting is that it prevents overfitting of camera model parameters for surveys with weak geometry. Agisoft Metashape allows alignment of the pictures with and without adaptive camera model fitting, and we found better results with adaptive camera model fitting enables. We prefer not to include such details in the manuscript (i.e., such information can be found in the Agisoft manual).

(R1.15) 89-99 Why are GCPs added between filter step 3 and 4 and not just after filtering?

Reply: We have followed the guidelines (http://www.rslab.se/agisoft-photoscan-pro/) of the Ljungbergslaboratoriet at the University of Umeå, Sweden. This guideline does in our experience leads to optimal DSM quality.

(R1.16) 100 Why is the orthophoto with sparse points used instead of dense? You can expect quite a lot of artefacts because the true 3D geometry is smoothed too strongly/not captured with enough resolution with the sparse point cloud.

Reply: Because of the irregular topography on the forested banks of the channel in this particular case it is better to use the smoothed sparse cloud to generate the orthophoto.

(R1.17) 103 What are the low noise points?

Reply: Photogrammetry point clouds are known to contain more noise (because of errors in photo matching, for example in shaded areas, vegetation, etc.) than lidar derived point clouds. When noise is present above the ground surface this is easily ignored by the ground finding algorithm, but as low points do not usually occur in lidar they are not accounted for and result in large dips in the ground surface. Therefore we remove them in a separate step based on a generalized ground surface using the 20th percentile point elevation and elimination of isolated points. This is now explained as follows in lines 105-111 of the manuscript: "The procedure removes low noise (i.e., noise below the actual ground surface) and filters overhanging vegetation, while retaining natural detail in the channel, and mostly avoids clipping at steep sections at the

channel banks and check dams. Low noise points are typical for dense point clouds generated using UAV photogrammetry, and were filtered by removing points more than 0.1 m below a smoothed 20th height percentile surface with a step size of 0.5 m. Overhanging vegetation was removed by classifying ground points using the lasground functionality in LAStools with 'ultra fine' settings. This setting effectively removes overhanging and sparse vegetation in the channel, but retains most of the fine details in the channel at the expense of including dense vegetation in geomorphologically inactive areas which were not of interest to our analysis.".

(R1.18) 106 How does the lasground tool work?

Reply: Lasground works based on "Progressive TIN densification" which is an iterative process of adding points to a TIN ground surface (through the lowest points until angle and step distance thresholds are reached (we refer to the LasTools manual for further details). It is an implementation of the algorithm presented in: Axelsson, P. (2000). DEM generation from laser scanner data using adaptive TIN models. International archives of photogrammetry and remote sensing, 33(4), 110-117.

(R1.19) 109 How was the point cloud rasterized (e.g. using IDW)?

Reply: Ground points are regularized into a grid format using a TIN interpolation to create a continuous surface. We now specify this in lines 112-113: "Finally, filtered points were rasterized into a DSM with a ground sampling distance of 5 cm using a TIN triangulation by LasTools.". We refer to the LasTools manual for further details.

(R1.20) 115 What is expected by increasing the number of surveys? The explanation of the reason for investigation is missing here. Do the authors expect an improvement due to even more potential tie points across more images?

Reply: This becomes clear further in the manuscript (section 4.2), where we show how a strong common geometry set by a few time steps forces time steps with less-strong geometry into the common geometry thereby greatly enhancing the relative accuracy.

(R1.21) 110-120 Are the GCPs measured in all images across time or only in one survey? Please, also assess the latter case because this could be a significant improvement regarding the time and potential availability of GCPs (e.g. maybe it is not possible to set them up each time).

Reply: The GCPs were measured in all the surveys, because this gives the cleanest comparison between the classical approach where all time steps are processed individually and the co-alignment approach. It is not the objective of this paper to evaluate whether it is possible to only apply GCPs to the first time step, and therefore we choose not to include this analysis.

(R1.22) 126 Thus, you evaluated the precision (if you compare the distances to the mean of the entire dataset) rather than the accuracy.

Reply: Indeed, but as explained above we have now also included an analysis of the absolute accuracy of the models.

(R1.23) 126-128 Please, also refer to the already existing literature discussing the error dependence to the distance to the GCPs. For instance, see Tonkin & Midgley (2016).

Reply: Done. We now cite this paper in line 175.

(R1.24) 128-129 I am not sure if this sentence is correctly stated because you do compare the point clouds directly using the mean of all nine point clouds at your selected points. Might you mean that you do not compare entire regions rather than just single points?

Reply: This is indeed what we mean. We have clarified this sentence as follows: "We did not compare entire point cloud regions because the channel changed substantially between events, and the banks were largely covered by forest with an irregular surface not suitable for accurate matching of the point clouds of the different surveys.".

(R1.25) 135-137 What is the difference between reprojection error and mean reprojection error? If the authors refer to the value provided by MetaShape it would be root

means squared reprojection error.

Reply: These are indeed the values reported by MetaShape. We have clarified this sentence as follows (lines 146-149: ". These surveys had relatively large root means squared reprojection errors and a relatively low number of tie points (Table 1). The root means squared reprojection errors of surveys 6 and 8 are 1.19 and 0.85 pix, respectively, while the root means squared reprojection error of the other surveys is on average 0.55 pix.".

(R1.26) 146-147 This finding is not new and well known in conventional photogramme-try as well as more recent work. Please, refer to these studies (e.g. Tonkin & Midgley (2016) and/or standard work (e.g. Kraus, 2007)).

Reply: We have included a reference to both papers.

(R1.27) 165-169 I do not understand this statement. How does a small (thus local) portion of the survey influence the entire/global change detection?

Reply: Figure 6 serves as an illustration of an omnipresent problem – this is thus not the only location where a large bias is observed. We have clarified this by including the following sentence: ". Such biases were common on the models of surveys 6 and 8 build via the classical approach.".

(R1.28) 170-171 As shown by Cook & Dietze (2019) and Feurer & Vinatier (2018)

Reply: These references have been included to this sentence.

(R1.29) 171-173 However, the comparison to independent references is still missing, which might indicate the more important aspect if the integration of GCPs does improve the absolute accuracy as well as the scaling of the combined model, which would be important for the volumetric change calculation.

Reply: As discussed above we have now also included an assessment of the absolute accuracy (see section 4.2).

[Figure]

(R1.30) 176-177 I might rephrase this sentence stating that the used co-alignment allows for more reliable change detection because during the reconstruction all images (from all surveys) are optimized within the same adjustment, using homologous image points covering several time-steps and therefore resulting in a joint camera geometry. Potential systematic errors are therefore spatially consistent.

Reply: Thanks for the suggestion, it makes our point indeed more clear. We have modified this sentence following the reviewer's suggestion (lines 206-210): "Co-alignment allows for more reliable change detection because during the reconstruction all images (from all surveys) are optimized within the same adjustment, using homologous image points covering several surveys and therefore resulting in a joint camera geometry. Potential systematic errors are therefore spatially consistent, and as a result do not influence comparisons between the 3D models of the surveys, such that their comparative accuracy is much higher and topographic change detection is more accurate.". 

Benjamin Purinton (Referee 2)

General Comments: (R2.1) The authors advocate for a modified method of change detection using UAV-SfM time series. Whereas the traditional approach uses each survey individually aligned to GCPs, the authors build upon and expand the work of Cook and Dietze (2019) in this journal to suggest a combination of GCPs and co-alignment over many (more than two) time steps. This manuscript is well written and well placed within the context of recent UAV-SfM studies. Despite being a short communication, the manuscript would benefit from some additional details, clarifications, and modifications that should all fall in the realm of minor revisions prior to acceptance.

Reply: We thank the reviewer for his positive evaluation of our manuscript and his constructive comments. We respond to the comments on a point-by-point basis below.

(R2.2) I understand that the authors lack a control dataset to test absolute accuracies, but I tend to agree with the other reviewer that some GCPs could be used as check points on absolute accuracy between the three approaches, listing (or better tabulating)

a simple metric like mean and standard deviation ought to suffice. Furthermore, the issue of GCPs is of great concern especially when surveying large areas where there may be difficulty in finding and reaching stable, clearly visible points to lay out targets and measure using dGPS. I am particularly interested in how increasing the number of GCPs (and their spatial arrangement in the area) affects the quality of CA+GCP method. Figure 3c,d go in this direction with plots showing increasing error with distance from GCPs. Interestingly the z error has the clearest trend with distance from GCP, whereas the xy error trend is less clear and only shows a decrease in accuracy for a few points at distances > _50 m. Would these trends be better presented in non-log space? It looks like a lot of the accuracies are sub-cm, so they may be "insignificant" (or at least negligible) for most geomorphic change detection studies, thus a linear (or semilog-x) plot would highlight the larger (1 cm - 1 m) inaccuracies that are of greater concern. If a log axis is preferred then the text should at least highlight and state the lower importance of these sub-cm inaccuracies. The xy trend may also be more apparent if less GCPs were used and the x-axis of that plot extended to some higher values (i.e., greater distances). If similar accuracy can be achieved with GCPs placed 80 m apart versus 20 m (e.g., accuracy _1 cm) then that's a significant improvement in field work time.

Reply: We have now included two extra scenarios, wherein we use 10 GCPs in the SfM and use the remaining (19) GCPs as check points. This analysis shows that co-alignment without GCPs leads to very low absolute accuracies despite having high relative accuracy. The absolute accuracies in xy and z directions are slightly higher when applying co-alignment than when processing each time step individually. The differences in absolute accuracies between these models are relatively low, however (see section 4.2 for further details). It is not our objective to evaluate the effects of the number of GCPs in this study, but rather focus on the accuracy of co-alignment compared to the classical approach where each survey is processed individually. Moreover, the effects of the number and distribution of GCPs has already been studied extensively for the classical approach (e.g., Kraus, 2011; Tonkin and Midgley, 2016). We have also

tried plotting Fig. 4c-d on linear or semilog-x axis, but this does actually makes the trends less clear. Moreover, the graph does not show any sub-cm accuracy points but rather sub-dm accuracy points (10-1 m = 0.1 m).

(R2.3) If the authors aim for reproducibility and standardization of UAV-SfM change detection then some additional details of the steps taken should be included. I have a couple suggestions here, which I mostly highlight in the line changes below. One point here: in the Section 3.2 Data processing, I suggest a nested alpha-numeric list (e.g., (1), (a), (b),...(2), (a), (b),...) rather than the steps in paragraph form. This will read more like a manual of the steps to follow. If these and previous suggestions are increasing the length significantly, I recommend removing Figure 2. I think the difference between these methods is quite clear in the text and the flow-chart is not necessary. Alternatively, the current Figure 2 could be replaced by a flow-chart of only the proposed method similar to Figure 2 in Cooke and Dietze (2019). This may negate the need for the alpha-numeric list, but that is for the authors to see and decide.

Reply: Personally, we find Fig. 2 clearer when it provides a side by side comparison of the three applied approaches. In addition, we do not favour a nested alpha-numeric list, as the prime objective of the manuscript is to show the differences in topographic change detection accuracy for difference UAV-SfM approaches. As such, we do not deem it necessary to present our methodology as a manual, especially as those readers interested in reproducing our steps can still clearly do this based on the presented text.

(R2.4) One missed study from this journal that adds some nice context is Duro et al. (2018). They used the traditional GCP approach without co-alignment on 8 surveys. It would be good to include this reference somewhere in the introduction or in the discussion as an example of the previously used (GCP only) technique for UAV-SfM surveys over many time steps (rather than just two). Table 2 and Figure 6 in that study could also provide some discussion comparisons with the absolute accuracies found using check points in the present study.

Reply: We follow the reviewer's suggestion and now cite Duro et al. (2018) in the introduction (line 42) and discussion (line 206).

(R2.5) A last general comment: Agisoft version 1.6 (in the present study version 1.5.2 is used) added a point confidence based on the number of depth maps a point appears in. I suggest the authors highlight this new feature to call attention to it. For instance this new feature could be used in further filtering of the point clouds to only extract high confidence points and/or as weights in interpolation schemes. There is no need to do any re-processing using the newer version, but this is certainly an important new feature that ought to be explored in future research. This could just be a few sentences in the discussion.

Reply: We agree that this is a very interesting new addition in Agisoft, from which we will surely benefit in the future. For this manuscript we have however done such post-processing with LasTools instead. Because we are not fully familiar with the new functionality in Agisoft yet we choose not to specifically mention the availability of this tool in the newest version of Metashape in the manuscript.

Line Changes: (R2.6) L33: "differentiated" to "differenced"

Reply: Corrected.

(R2.7) L45: should "reduces the accuracy" instead say "increases the accuracy"?

Reply: Yes indeed! We have corrected the sentence.

(R2.8) L46-47: I think the terms relative and absolute accuracy (as opposed to comparative and external) are more common. It is fine to continue using comparative and external but maybe quickly define, e.g., "comparative – or relative – accuracy... external – or absolute – accuracy"

Reply: Done.

(R2.9) L57: remove hyphen from "common-practice"

Reply: Done.

(R2.10) L94: Please include the specs of the computer used (e.g., RAM, number of CPUs, presence or absence of a GPU) and approximate processing times (this could come at the end of the paragraph). I'm especially curious about how long the 9 survey block co-alignment took on whatever computer was used.

Reply: We have run Agisoft Metashape on the high performance computing (HPC) cluster of our faculty. We typically used five nodes, which each have two AMD EPYC 7451 24-Core Processors and 256 GB memory, and are equipped with a GeForce GTX 1080 Ti graphics card. The number of nodes actively used depended on their availability (depending on the activity on the HPC) and therefore we cannot make a sensible comparison of processing times here. Nevertheless, with this device co-alignment of the 9 surveys could be finished (from initial alignment to final dense cloud and orthophoto) within a few hours (of course excluding the time of manual GCP placement).

(R2.11) L95: Were these steps done using the Agisoft Python API or strictly through the Agisoft GUI? If Python, consider including a GitHub repository with the code. Or at least a statement saying whether the API or GUI was used. The iterative steps of tie point removal and re-alignment would be especially ideal for a Python script. As earth surface processes research moves towards increasing reproducibility the move to scripting as opposed to clicking steps is vital, or at least more manual-like details (e.g., the alphanumeric list or detailed flow-chart I suggest in general comment).

Reply: We have used a combination of batch run functionality of the Agisoft GUI and Python API. The iterative steps of tie point removal and re-alignment do call on a few lines of code using the Python API, but only use operations which are commonly available through the GUI. The script consists of a few lines only and is therefore too thin for publication on GitHub, but we are happy to share it upon request. While full automation and reproducibility are useful when running many similar scenarios, visual inspection of intermediate results do have their role in ensuring sensible results when processing

new or more variable datasets favouring the use of a GUI.

(R2.12) L98-99: All GCPs were added between steps 3 and 4? Or just a few and then kept adding more with each iteration? This could be the point where 10, 25, 50, 75, 90% (or so) of the GCPs are added each time to see the improvement / changes in accuracy.

Reply: As discussed above, we have mostly worked with all GCPs but have now also included a scenario where we use 10 GCPs for DSM construction and the remaining GCPs as check points to evaluate the absolute model accuracy. It is not our objective to evaluate the effects of the number of GCPs in this study, but rather focus on the accuracy of co-alignment compared to the classical approach where each survey is processed individually. Moreover, the effects of the number and distribution of GCPs has already been studied extensively for the classical approach (e.g., Kraus, 2011; Tonkin and Midgley, 2016).

(R2.13) L101 to end of paragraph: include a few more details on the lastools commands used (e.g., lasground, what were all the parameters?), and also include details of the "rasterization" scheme. Here "rasterized" should be changed to "interpolated" and the method of interpolation should be stated (e.g., IDW?).

Reply: Ground points are regularized into a raster format using a TIN interpolation to create a continuous surface. We now specify this in lines 112-113: "Finally, filtered points were rasterized into a DSM with a ground sampling distance of 5 cm using a TIN triangulation by LasTools.". We refer to the LasTools manual for further details.

(R2.14) L103: Should "low noise" say "high noise"? Or maybe just "low accuracy".

Reply: We do actually mean to indicate noise points that are low elevation (below the actual ground surface) – which we now specify in line 105 of the manuscript. Photogrammetry point clouds are known to contain more noise (because of errors in photo matching, for example in shaded areas, vegetation, etc.) than lidar derived point

clouds. When noise is present above the ground surface this is easily ignored by the ground finding algorithm, but as low points (low noise) do not usually occur in lidar they are not accounted for and result in large dips in the ground surface. Therefore, we remove them in a separate step based on a generalized ground surface using the 20th percentile point elevation and elimination of isolated points.

(R2.15) L113: "changed" should be "change"

Reply: Corrected.

(R2.16) Figure 3: What are the error bars in a and b? I suggest removing the median bars and just use the mean, since the median is never referenced in the text and Figure 4 uses the mean. However, the median error bars show large overlap, which is concerning and would seem to limit the interpretation of significant improvements. Do the authors have a response to this concern?

Reply: Error bars denote standard deviation for the mean and 25th and 75th percentile for the median – we now specify this in the captions of Figs. 3 and 4. We prefer show both mean and median in the Figs. 3 and 4, as it gives the reader more information than just the mean. To mitigate the error bars in our analyses we have now performed t-tests to evaluate whether the means of the tested approaches are significantly different. Results from the t-tests are specific in the manuscript where appropriate (see sections 4.1 and 4.2).

(R2.17) Figure 3: In the caption "accuracy decreases with distance" should perhaps be "accuracy decreases with increasing distance"?

Reply: Yes. Corrected.

(R2.18) L147: In the xy direction this decrease in accuracy is only apparent in the few datapoints > _50 m from a GCP (if one considers < _1 cm inaccuracies as negligible in this context). Be more specific about these distance breaks in the text, or perhaps use the suggested gradual addition of GCPs to highlight this.

Reply: As specified above there are no sub-cm inaccuracies, only sub-dm inaccuracies (10-1 = 0.1 m).

(R2.19) Also, what are the fitted lines in Figure 3c,d? Is this a moving average? Please state this.

Reply: The fitted lines are indeed moving averages. This has now been specified in the caption of Fig. 3 (now 4): "The fitted lines represent a moving average.".

(R2.20) L146-160: In the results please reference Figures 3 and 4 where appropriate.

Reply: Done.

(R2.21) L163: The "large errors" in volume change don't refer to any reference data. I'm not sure it's appropriate to refer to the volume change errors without a control (e.g., lidar). Instead these are relatively (compared with the other alignment techniques) large changes in volume that are outside of expectations. Based on Figure 6, it seems that this error is a visual assessment, which is fine but should be clarified.

Reply: We have used 48 validation point for which we knew for sure that they didn't change over the nine surveys. As such, this is not simply a visual assessment. In the example in Fig. 6 we for example know for sure that the check dam didn't change in elevation.

(R2.22) Figure 6: It would be helpful to just put a colorbar on this and remove the caption description (+/-4 m in color and +/-0.25 m transparency).

Reply: To ensure maximum visibility, we prefer a "clean" version of this figure. Moreover, the exact values of the change are only of minor importance for the main message of this figure.

(R2.23) L198: "has" should be "have"

Reply: "Has" is correct. 

Anonymous Referee #3

(R3.1) While is gratifying that the workflow we discussed in Cook and Dietze, 2019 (and was proposed by Feurer and Vinatier, 2018) was shown to be effective even when GCPs are used, I found this paper very thin for a stand-alone paper, even a short communication – it feels like something carved out as a least publishable unit when it could be included in a paper that discusses the observed changes (I notice that the authors appear to have such a manuscript in review). While the additional testing of the method is nice, it's not clear that this manuscript adds much new knowledge or methodological development. This is perhaps best exemplified by the discussion, much of which (lines 176-187, 191-199, 204-208) restates points that were already made in Cook and Dietze. The authors state one the key unanswered questions they address is "how well coalignment does on a larger number of surveys" and that Cook and Dietze tested only two surveys. This is not the case; while most of the results in Cook and Dietze were from survey pairs, we also applied the method to a set of 4 surveys (shown in Fig. 6 in our paper) and note that including additional surveys could improve results in some cases. While 9 is greater than 4, it's not clear that this is enough of a difference to be noteworthy. This leaves the addition of accurate GCPs as the only new thing (note that Feurer and Vinatier also used GCPS, just not precise ones). Overall, I am doubtful that this manuscript makes a significant scientific contribution, again best illustrated by the lack of original ideas in the discussion.

Reply: Indeed, our paper is a clear follow-up on the work presented in Feurier and Vinatier (2018) and Cook and Dietze (2019). As a logical result, some of the findings that we present in our manuscript are line with those of Cook and Dietze (2019). Nevertheless, we do present a range of findings that are novel and which will be of interest to the scientific community, including: - The performance of UAV-SfM co-alignment including GCPs. We demonstrate that compared to the classical approach wherein surveys are processed individually co-alignment enhances the accuracy of topographic change detection by a factor 4 with GCPs and a factor 3 without GCPs, leading to xy and z offsets < 0.1 m for both co-alignment approaches. Note that Feurier and Vinatier did use low accuracy GCPs but did not use UAV imagery, which is currently one of the most used techniques for DSM generation. - Quantification of absolute errors. We now quantify both the absolute and relative errors associated to the three UAV-SfM approaches studied here. - Co-alignment may outperform the classical approach. While Cook and Dietze (2019) suggested that co-alignment without GCPs can be used for change detection with a level of detection comparable to that of a survey grade GCP-constrained pair of models, for our dataset combining nine surveys, co-alignment without GCPs outperforms the classical approach where surveys are processed individually with GCPs. - Application to large, multi-survey, datasets. We showcase that co-alignment can be applied to large multi-survey datasets – in this case 9 surveys. After a careful re-read we realize that Cook and Dietze did indeed also applied co-alignment to 4 surveys in one case, and we have now removed related incorrect claims from our manuscript. Still, we argue that we provide more detailed and new insights into co-alignment of multi-survey datasets, as indicated below. - Showcasing increased accuracy of poorly aligned surveys. We show how co-alignment leads to particularly large improvements in the accuracy of poorly aligned surveys with few tie-points and large reprojection errors that have severe offsets when processed individually, by forcing them onto the more accurate common geometry set by the other surveys. - Distance to nearest GCP. We provide insight into how the relative accuracy in the xy and z directions increases with distance from the nearest GCP.

In short, it is incorrect to classify our manuscript as "a least publishable unit", as it presents a number of novel findings. In addition, our "In Review" paper has now been published (https://doi.org/10.1038/s41598-020-71016-8), and in this paper we do not apply co-alignment and also analyse a different reach of the Illgraben channel.